# Research on Carbon Emissions Prediction Model of Thermal Power Plant Based on SSA-LSTM Algorithm with Boiler Feed Water Influencing Factors

**Xindong Wang [1], Chun Yan [1,\*], Wei Liu [2,\*] and Xinhong Liu [3]**

1    College of Mathematics and Systems Science, Shandong University of Science and Technology, Qingdao 266590, China
2    College of Computer Science and Engineering, Shandong University of Science and Technology, Qingdao 266590, China
3    Beijing Institute of Petro-Chemical Technology, Beijing 102617, China
\*    Correspondence: yanchun@sdkd.net (C.Y.); liuwei@sdust.edu.cn (W.L.)

**Abstract:** China's power industry is a major energy consumer, with the carbon dioxide ($CO_2$) generated by coal consumption making the power industry one of the key emission sectors. Therefore, it is crucial to explore energy conservation and emissions reduction strategies suitable for China's current situation. Taking a typical cogeneration enterprise in North China as an example, this paper aims to establish a generalized regression prediction model for carbon emissions of coal-fired power plants, which will provide a reference for China to seek strategies for carbon peaking and carbon neutralization in the future. Firstly, in terms of the selection of influencing factors, this paper uses objective index screening methods, simulation means, and the eXtreme Gradient Boosting algorithm (XG-Boost) to analyze the feature importance of various influencing factors. It is concluded that the relevant influencing factors of the boiler feed water system have a strong correlation and characteristic importance with the carbon emissions results of coal-fired power plants. Therefore, this paper proposes to introduce these factors into the regression prediction model as auxiliary variables to more scientifically reflect the carbon emissions results of coal-fired power plants. Secondly, in the aspect of regression prediction model establishment, inspired by the sparrow's foraging behavior and anti-predation behavior, this paper selects the sparrow search algorithm (SSA) with strong optimization ability and fast convergence speed to optimize the super parameters of the long short-term memory network algorithm (LSTM). It is proposed to use the SSA-LSTM algorithm to establish the carbon emissions regression prediction model of coal-fired power plants. The advantage of the SSA-LSTM algorithm is that it can effectively simplify the super parameter selection process of the LSTM algorithm, effectively solve the global optimization problem, prevent the model from falling into overfitting and local optimization, and make the carbon emissions regression prediction model of coal-fired power plants achieve a better fitting effect. By comparing the performance indicators of the model before and after the improvement, it is found that the regression prediction effect of the SSA-LSTM coal-fired power plant carbon emissions regression prediction model, which introduces boiler feed water influencing factors, has been effectively improved. Therefore, the model proposed in this paper can be used to conduct a comprehensive impact factor analysis and regression prediction analysis on the carbon emissions intensity of China's coal-fired power plants, formulate targeted carbon emissions reduction countermeasures, and provide a theoretical basis for energy conservation and emissions reduction of China's coal-fired power plants.

**Keywords:** carbon emissions; SSA algorithm; LSTM algorithm; XG-Boost algorithm; boiler feed water system

## 1. Introduction

In 2020, COVID-19 was prevalent worldwide, reducing carbon dioxide emissions in the short term, which proves that human beings are fully capable of improving air quality in the short term. According to statistics, because a large number of people around the world were required to work at home and international borders were closed, the consumption rate and transportation rate decreased accordingly. Based on the estimation of the International Energy Agency (EEA), the global energy consumption decreased by at least 6% in 2020, which is approximately equal to the annual energy consumption of India. The report of CarbonBrief, a British research institution, estimated that global carbon emissions will be reduced by 20–30 billion tons this year, resulting in a reduction of about six times that after the global financial crisis in 2009, and the total carbon emissions reduction will be approximately twice that caused by all crises since the end of World War II, including the Middle East oil crisis and the financial crisis. Looking at human crises in history, the economic recovery is likely to bring a retaliatory rebound. If appropriate policies are not taken in time, it is likely to lead to a rapid growth of carbon emissions. After the global COVID-19 pandemic, facing the situation that the economy has been greatly impacted, countries may take various measures to stimulate their own economies, which is likely to cause an increase in carbon emissions in a short time. Therefore, it is crucial for countries to seek solutions that take into account both carbon emissions reduction and economic recovery.

The consumption of fossil energy is the main source of carbon emissions. Since 2014, China's energy conservation and emissions reduction have achieved initial results, but China's energy utilization rate is low and the energy consumption structure of the power sector is dominated by coal consumption, with a low proportion of renewable energy, which has caused great damage to the climate and the environment. Therefore, for China, energy conservation and emissions reduction in the power sector is a long-term and arduous task, an important prerequisite for China's sustainable development, and the only way to curb global warming. Through the analysis of China's energy, form, and environmental situation, it can be seen that the energy consumption of the power sector has a crucial impact on carbon dioxide emissions. In the context of global energy conservation and emissions reduction, the analysis of factors affecting China's energy consumption is conducive to a comprehensive understanding of the current situation of carbon emissions, while the research on emissions prediction can have a basic understanding and grasp of the future trend of carbon emissions, which is of great practical significance.

The coal-fired power generation industry is one of the major carbon emissions sources in China, with its coal consumption accounting for more than 50% of the total national coal consumption [1]. At present, China is gradually conducting carbon market research, and the implementation of carbon trading and carbon tax policy is a general trend [2–4]. In order to reduce the carbon emissions cost of the coal-fired power generation industry and improve its competitiveness, it is necessary to accelerate the research and application of carbon emissions reduction technology of coal-fired power plants. The use of circulating fluidized bed (CFB) boiler units is an important way to solve the problem of using a large quantity of low-quality coal in China, and also an important way to achieve clean coal combustion [5,6]. CFB boilers can not only burn low-quality coal with low calorific value, but also achieve low-cost emissions reduction of NOx and $SO_2$ through low-temperature combustion in furnace and limestone calcination [7–9]. In view of the above advantages, CFB boilers have been widely used in China, especially in low-quality coal production areas such as Shanxi, Shaanxi, and Inner Mongolia [10]. By the end of 2018, the installed capacity of CFB boiler units in China had exceeded 82.3 million kW, accounting for approximately 8.16% of the total installed capacity of coal power [11]. Therefore, the purpose of this research paper is to decompose and forecast the influencing factors of China's coal-fired power generation industry to form a complete research system, explore the change characteristics and possible development trends of China's coal-fired power generation industry carbon

emissions, and provide a reference for China to formulate development plans, promote energy structure optimization, and industrial upgrading in the future.

## 2. Literature Review

To formulate targeted emissions reduction policies and achieve the expected results, China must understand the factors that lead to the increase of carbon dioxide emissions, in order to effectively reduce carbon dioxide emissions by controlling these factors. Many scholars have studied the influencing factors of carbon dioxide emissions. Dong Feng [12] applied the LMDI method to decompose the influencing factors of carbon emissions into economic scale, industrial structure, technological progress, and energy structure, and found that economic scale is the main driving factor of carbon emissions growth, while technological progress plays a negative role. Based on the LMDI factor decomposition method, Zhou Yannan [13] analyzed the changes of carbon emissions in different provinces and regions from three aspects: economic scale, structural transformation, and technological progress. The study showed that the leading effects of carbon emissions in different periods varied between different provinces and regions. Yan Qingyou [14] used the GDIM model to study the driving factors of $CO_2$ emissions change of thermal power generation in China from 2000 to 2016. The decomposition results show that economic activities, power demand, and energy use play a positive role in driving carbon emissions of thermal power generation in China. Gao Sihui et al. [15] and Liu Xiaodie [16] used a Lasso regression model to screen out eight important carbon emissions influencing factors, simulated and set the value of each influencing factor from 2019 to 2030, and established a BP neural network model to predict the carbon emissions and peak time of Jiangsu Province from 2019 to 2030. Therefore, in terms of analysis of the influencing factors of carbon emissions from energy consumption, there are many mature research methods, and there is no final conclusion on which method can accurately analyze the influencing factors. However, based on a range of research, combined with the empirical research and comparative analysis of the decomposition methods of carbon emissions by experts and scholars in the energy field in recent years, the XG-Boost method selected in this paper can achieve the desired effect on the decomposition of the factors affecting carbon emissions.

Global warming has attracted increasing attention, and scholars have made rich research achievements in carbon emissions prediction. At present, the research mainly focuses on the prediction of carbon emissions and their peaks. Scholars have studied carbon emissions using different types of models based on different theoretical bases and different research perspectives. Shi Shaoqing [17] and others used the multi-resolution time series neural network (MTNN) model to achieve the power consumption prediction task. Sun Xiujuan [18] applied the information entropy theory to traffic flow forecasting, and established a variable weight combination forecasting model based on the entropy weight method by using the hidden information contained in the objective data. Dong Xiaogang [19] used the DTGARCH Laplace model to forecast the return on assets, providing investors with a more accurate decision-making basis. Duan Fumei [20] used the improved BP neural network model to predict peak carbon emissions in China under eight development models. The research shows that China can achieve the goal of reaching peak carbon emissions by 2030 under the economic recession model, energy-saving model, and other models. Wang Hehe [21] used the whale optimization algorithm to improve the limit learning machine model, overcoming the deficiency of the limit learning machine that is easy to fall into the local optimal solution, and used the improved model to predict China's carbon emissions and intensity from 2019 to 2040, accurately reflecting China's future carbon emissions trend. Chiun SinLin [22] used the gray prediction model to predict the trend of Taiwan's carbon dioxide emissions from 2010 to 2012. Cao Zhengsheng [23] used the autoregressive integral moving average ARIMA prediction model to predict carbon emissions, and calculated the regression results and predicted values using temperature and global greenhouse gas emissions. After calculation, they completed 16% carbon emissions reduction annually, and predicted the global average temperature to increase by 1.6 °C by 2050, reaching the

goal set by the United Nations. Zhao Jinyuan [24] established a BP neural network and multiple linear regression model to predict and analyze carbon emissions of iron and steel enterprises, and found that the BP neural network model is superior to the multiple linear regression model in predicting carbon emissions. Hu Jianbo [25] deduced and predicted the change trend of China's carbon emissions intensity based on a LSTM neural network model and certain economic growth expectations. At the same time, they established an ARIMA-BP neural network model as a verification model to directly predict the carbon emissions intensity, which verified the robustness and accuracy of the prediction model. Yaqin Hu [26] estimated the $CO_2$ emissions of large-scale coal-fired power plants using Orbiting Carbon Observatory-2 (OCO-2) satellite data based on remote sensing inversions and bottom-up methods, Machine learning algorithms extract key information and the main data features from known datasets to make accurate identification and prediction of future data. Tong Xin [27] used the gray model GM(1,1) to predict China's carbon emissions from 2012 to 2020. According to the predicted results, China still faces great pressure to reduce emissions. Meng [28] proposed a new prediction model based on the GM(1,1) model by introducing a linear equation, and predicted China's carbon emissions in the past 20 years. The results showed that the improved model had a better prediction effect. Song [29] proposed an improved gray multivariate model, which avoids the inaccuracy and poor adaptability of the traditional gray model, and used the improved model to predict China's carbon emissions, providing reasonable guidance for the formulation of energy consumption plans. Liu [30] combined the gray model with the autoregressive comprehensive moving average model and the second-order polynomial regression model, and optimized the model parameters by using the PSO method. The results show that if the government did not control carbon emissions, it was difficult for China to achieve the set goal in 2020.The gray prediction model is simple in application, with few parameters, is easy to train, and has high prediction accuracy. However, the fitting ability of the model is inferior to that of the machine learning algorithm. A large number of studies have shown that machine learning can better deal with prediction problems, and also proved that the idea of applying machine learning to prediction in the energy field is feasible.

In summary, in the relevant literature on carbon emissions of coal-fired power plants, there are many studies on the establishment of carbon emissions prediction models of coal-fired power plants using traditional algorithms. The selection of influencing factors for predicting the carbon emissions of coal-fired power plants mostly focuses on direct carbon emissions influencing factors, and there are few studies on the optimization and improvement of models and the dimensional expansion of influencing factors. In terms of the selection of influencing factors, this paper uses the XG-Boost algorithm to analyze the feature importance of influencing factors on the basis of statistical analysis. Compared with principal component analysis, analytic hierarchy process, and other subjective methods, the XG-Boost algorithm can objectively analyze various influencing factors of carbon emissions of coal-fired power plants, making the selection of influencing factors more scientific. In terms of establishing the prediction model, inspired by the sparrow's foraging and anti-predation behavior, this paper selects the sparrow search algorithm (SSA) with its strong optimization ability and fast convergence speed to optimize the super parameters of the LSTM algorithm, and proposes to use the SSA-LSTM algorithm to establish the carbon emissions regression prediction model of coal-fired power plants. The advantage of the SSA-LSTM algorithm is that it can effectively simplify the super parameter selection process of the LSTM algorithm, effectively solve the global optimization problem, prevent the model from falling into overfitting and local optimization, and enable the carbon emissions regression prediction model of coal-fired power plants to achieve a better fitting effect. Through multiple screenings of influencing factors, this paper determines that the relevant influencing factors of the boiler feed water system have a strong correlation and characteristic importance with the carbon emissions results of coal-fired power plants. While previous studies have ignored this more important influencing factor, this paper proposes to introduce it as an auxiliary variable into the model, which can more reasonably

reflect the carbon emissions results of coal-fired power plants, and then introduce the SSA-LSTM algorithm, The optimized carbon emissions regression prediction model is established to improve the prediction performance of the model on carbon emissions of coal-fired power plants.

## 3. Algorithm Principle

### 3.1. XG-Boost Algorithm Flow

Starting from decision tree depth 0, each node traverses all the features, sorts the features according to the values in the features, and then scans the features linearly to determine the best segmentation point. Finally, after all the features are segmented, the feature with the highest Gain is selected. The calculation of Gain is shown below.

$$Obj = -\frac{1}{2}\Sigma_{j=1}^{T}\frac{G_j^2}{H_j + \lambda} + \gamma \tag{1}$$

where two re parameters $\lambda$ and $\gamma$ are set, $T$ is the number of leaf nodes, and part $G/(H + \lambda)$ of the objective function of Formula (1) represents the contribution degree of each leaf node to the current model loss. After fusion, the calculation expression of Gain can be obtained, as shown in Formula (2).

$$\text{Gain} = \frac{1}{2}\left[\frac{G_L^2}{H_L + \lambda} + \frac{G_R^2}{H_R + \lambda} - \frac{(G_L + G_R)^2}{H_L + H_R + \lambda}\right] - \gamma \tag{2}$$

At the node, the sample is divided into two sets of left child node and right child node, the $G_L$, $H_L$, $G_R$, and $H_R$ values of the two sets are calculated, respectively, and the Gain is then calculated. According to the superposition of feature Gain, to judge the Gain of a feature, it is necessary to obtain its corresponding Gain in each tree, respectively, and then calculate the Average Gain corresponding to each feature. You can obtain a measure of how important a feature is to the whole model in terms of the Gain coefficient.

### 3.2. Short- and Long-Term Memory Network LSTM

The traditional artificial neural network, namely, the Feed Forward Neural Network (FNN), has no memory function and can only use the information of the current moment when processing sequence data, ignoring the role of historical information. In the Recurrent Neural Network (RNN), however, hidden layer units act as memory modules and can be constantly updated as new data are entered. However, if a sequence is long enough, it will be difficult for them to transmit information from the earlier time step to the later time step, which means that RNNS have only short-term memory.

LSTM introduces the short and long-time memory module into the recurrent neural network, which is a special recurrent neural network that can capture the long-time dependence and effectively utilize the long-distance sequence information. At present, the LSTM network is the most widely used, as shown in Figure 1. The LSTM unit replaces the hidden layer in the original RNN network with a chain structure.

The LSTM structure shown above is composed of four neural network layers, where $x_t$ represents the network input, $h_t$ represents the output, and $\sigma$ represents the sigmoid function. The LSTM unit specially designs memory cells to store historical information. The current state of memory cells is represented by $C_t$, which extends straight along the whole chain. The update and protection of the memory cell state information is controlled by the gate structure, which is a way for information to pass selectively. They consist of a sigmoid neural network layer and a dot product operation. The sigmoid layer outputs data between 0 and 1, indicating how much information each component should pass. A value of 0 means that no information can pass, while a value of 1 means that all information can pass. In the LSTM structure, the gate structure can be divided into the input gate, forgetting gate, and output gate.

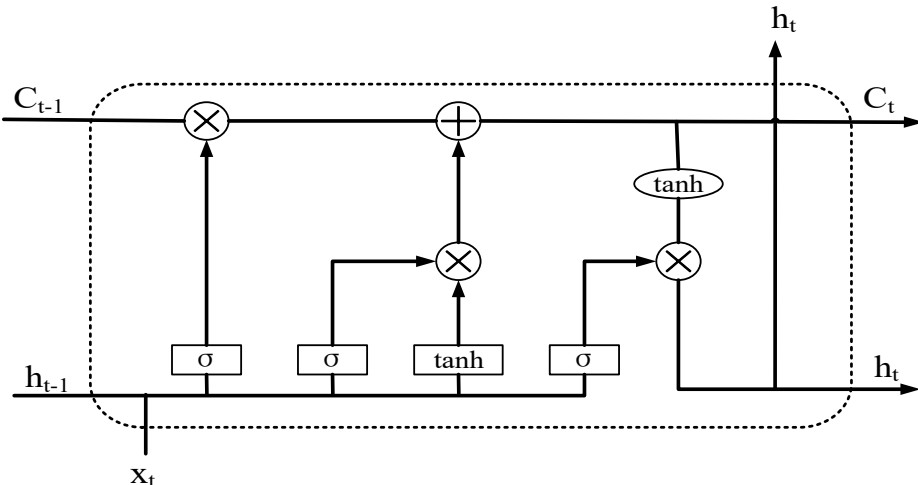

**Figure 1.** LSTM structure diagram.

Forgetting gate: Information that is no longer useful in the memory cell state can be discarded through the forgetting gate. Input $x_t$ and $h_{t-1}$ are sent to the forgetting gate, and the binary output $f_t$ is obtained after passing through the sigmoid activation function layer. It is then multiplied point by point with the memory cell state $C_{t-1}$. If the value of $f_t$ is 0, the corresponding information is discarded; otherwise, the corresponding information is reserved for future use. Thus, Equation (3) can be obtained.

$$f_t = \sigma\left(W_f[h_{t-1}, x_t] + b_f\right) \tag{3}$$

where $W_f$ represents weight matrix and $b_f$ represents bias.

Input gate: Adding useful information to the state of the memory unit is determined by the input gate, with the specific realization process as follows: Firstly, the sigmoid function is used to adjust the information, and similar to the function of the forgetting gate, input $x_t$ and $h_{t-1}$ are used to filter the value $i_t$ to be remembered. Vector $\widetilde{C}_t$ of the candidate values is then created using the tanh function, which can give an output ranging from −1 to +1 with all possible values from $x_t$ and $h_{t-1}$. Finally, the two parts are combined and the value of the vector is multiplied by the adjustment value to obtain useful information. Thus, Equations (4) and (5) can be obtained.

$$i_t = \sigma(W_i[h_{t-1}, x_t] + b_i) \tag{4}$$

$$\widetilde{C}_t = \tanh(W_c[h_{t-1}, x_t] + b_c) \tag{5}$$

After the control process of the forgetting gate and input gate, the updating formula of the memory unit state can be given as Equation (6).

$$C_t = f_t \cdot C_{t-1} + i_t \cdot \widetilde{C}_t \tag{6}$$

Output gate: Extract useful information from the current cell state to render for the output of tasks performed by the output gate. First of all, run a sigmoid layer to decide which information contributes to the state of the output cell, and also to the cells by tanh, after they are both the result of the multiplication. The final output value will be obtained, and sent as input to the next unit, namely, Equation (7).

$$h_t = \sigma(W_o[h_{t-1}, x_t] + b_o) \cdot \tanh(C_t) \tag{7}$$

In summary, the design of three gates and an independent memory unit enables the LSTM unit to store, read, reset, and update historical information over long distances.

### 3.3. Sparrow Optimization Algorithm

Inspired by the foraging and anti-predation behaviors of sparrows, researchers proposed an optimization algorithm with strong searching ability and fast convergence, called the Sparrow Search Algorithm (SSA).

The effectiveness of the SSA is evaluated for common optimization problems in engineering applications, and it is often used to solve global optimization problems due to its advantages of simplicity, flexibility, and efficiency. Similar to the division of labor in sparrow populations, SSA is divided into three stages: discoverer, follower, and investigator.

Discoverers find food and provide directions for other individuals in the crowd, so discoverers look for a wide variety of food. They comprise 20% of the population. The discoverer's position update formula is as follows:

$$\overline{X}_{i,j}^{t+1} = \begin{cases} \overline{X}_{i,j}^t \cdot \exp\left(\frac{-h_s}{\alpha \cdot M_s}\right), & if \ R_s < ST \\ \overline{X}_{i,j}^t + Q \cdot L, & if \ R_s \geq ST \end{cases} \tag{8}$$

In Equation (8), $h_s$ represents the current iteration number; $M_s$ represents the maximum iteration number; $\overline{X}_{i,j}$ represents the current position of the ith sparrow in the jth dimension; $\alpha \in [0,1]$ represents the random number; and $R_s$ and $ST$ represent the warning value and safety value, respectively. When $R_s \in [0,1]$ and $ST \in [0.5,1]$, $Q$ is the normal distribution of the random number, $L$ represents 1, and all elements are $1 \times D$. When $R_s < ST$, it means that the community environment is safe and no predators are found in the vicinity, and the discoverer can conduct a wide-area search mechanism. When $R_s \geq ST$, it means that the individuals in the group have found the predator and have sounded the alarm. All the individuals in the group will take anti-predator action and the discoverer will lead the followers to the safe position.

In SSA, because discoverers are responsible for finding food for the whole population and providing foraging directions, they can usually obtain a larger foraging search area. Discoverers with a high fitness value will obtain food first in the search process.

Followers perform a food search after searching with the discoverer and the community around the discoverer's location. The position update formula of followers is as follows:

$$\overline{X}_{i,j}^{t+1} = \begin{cases} Q \cdot \exp\left(\dfrac{\overline{X}_{worst}^t - \overline{X}_{i,j}^t}{i^2}\right), if \ i > \frac{n}{2} \\ \overline{X}_p^{t+1} + \left|\overline{X}_{i,j}^t - \overline{X}_p^{t+1}\right| \cdot A^+ \cdot L, other \end{cases} \tag{9}$$

In Equation (9), $\overline{X}_p$ is the optimal position currently occupied by the discoverer; $\overline{X}_{worst}$ represents the worst position currently occupied; and $A$ is the matrix of $1 \times D$, where the value of each element is 1 or $-1$ and $A^+ = A^T(AA^T)^{-1}$. When $i > n/2$, the sparrow population senses danger and fights back.

Investigators are individuals randomly selected from the population. When predators invade, they send a signal to the sparrow to escape to a safe place. The behavior formula of the investigator is as follows:

$$\overline{X}_{i,j}^{t+1} = \begin{cases} \overline{X}_{best}^t + \beta \cdot \left|\overline{X}_{i,j}^t - \overline{X}_{best}^t\right|, & if \ f_i \neq f_g \\ \overline{X}_{i,j}^t + K \cdot \left(\dfrac{\left|\overline{X}_{i,j}^t - \overline{X}_{worst}^t\right|}{(f_i - f_w) + \varepsilon}\right), & if \ f_i = f \end{cases} \tag{10}$$

In Equation (10), $\overline{X}_{best}$ is the current global optimal position; $\beta$ is the control step parameter and is a normal distribution random number with mean 0 and variance 1; $K \in [-1,1]$ is a random number, represents the direction of sparrow movement, and can also control the movement step; $f_i$ is the fitness value of the sparrow, with $f_g$ and $f_w$ the best

and worst fitness values in the current search range, respectively; and $\varepsilon$ is the smallest real number to prevent the denominator from being 0. When $f_i \neq f_g$, it means that the sparrow is at the boundary of the population and vulnerable to attack by predators, so it needs to adjust its position. When $f_i = f_g$, it indicates that individual sparrows in the population are aware of the danger and need to group with other sparrows to avoid the danger.

## 4. Data Introduction

### 4.1. Data Source

This paper selects a large-scale cogeneration enterprise in North China as the data source. The thermal power plant is a subsidiary company of a group Co. Ltd. It is an enterprise-owned thermal power plant, which mainly undertakes the heating and power supply tasks of the group and surrounding units and residents. It is a large, modern cogeneration enterprise integrating power generation and steam supply. The production operation diagram and production operation layout are shown in Figures 2 and 3, respectively.

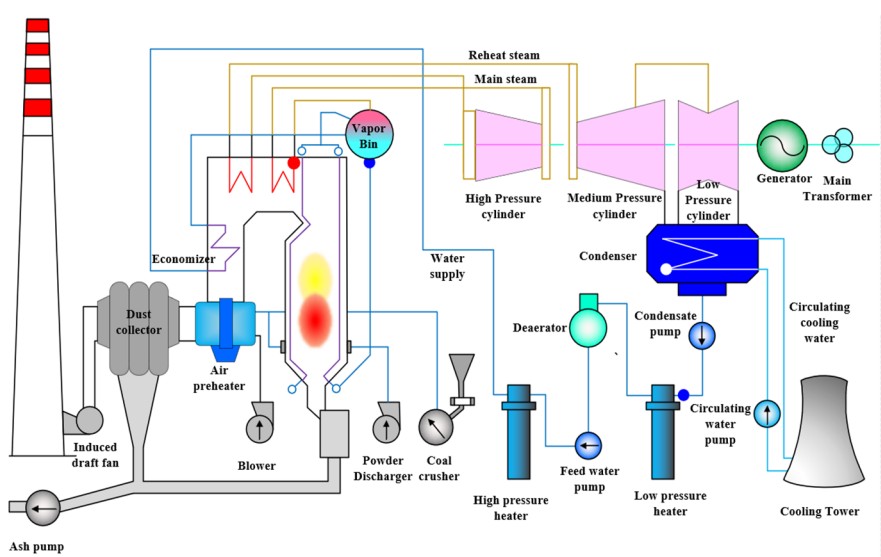

**Figure 2.** Schematic diagram of production operation of cogeneration enterprise.

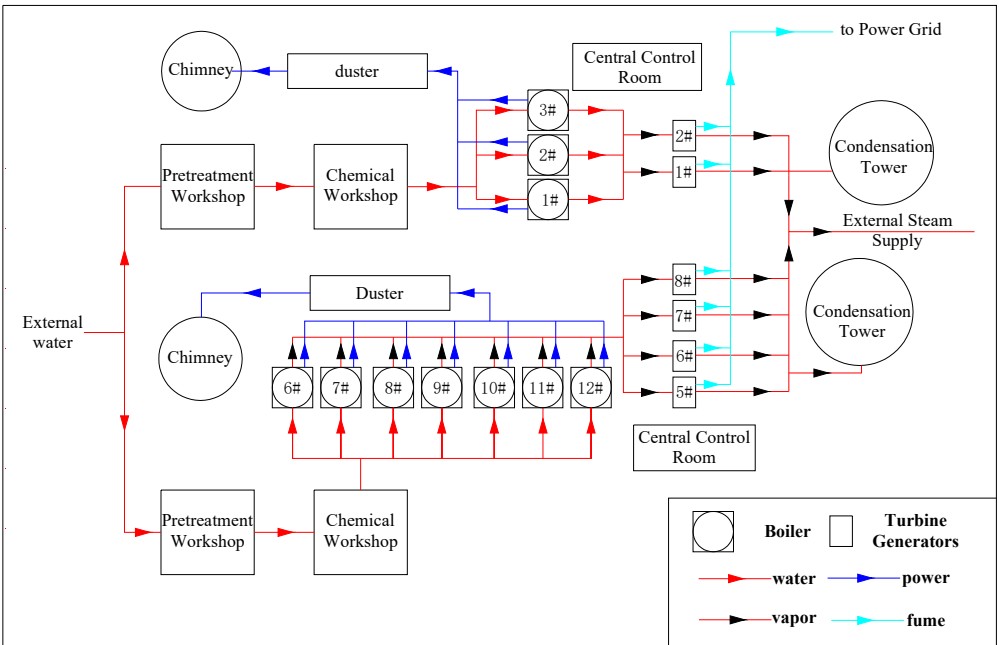

**Figure 3.** Layout of production operations of cogeneration enterprises.

The company has both the first and second thermal power plant. Until now, the company has had ten boilers in operation, of which the first thermal power plant has one 220 t/h circulating fluidized bed (CFB) boiler and six 240 t/h circulating fluidized bed boilers, while the second thermal power plant has three 600 t/h circulating fluidized bed boilers, with a total steam production capacity of 3460 t/h. There are five units in operation, including one 24 MW condensate extraction unit, one 24 MW back extraction unit, and one 30 MW back extraction unit in the first thermal power plant. The second thermal power plant has one 150 MW condensate extraction unit and one 30 MW back extraction unit, with a total installed capacity of 258 MW. In normal operation mode, five boilers at the first thermal power plant are in operation and two boilers are on standby. The second thermal power plant has two boilers in operation and one on standby. Five units are in operation. The structure of the circulating fluidized bed boiler is shown in Figure 4.

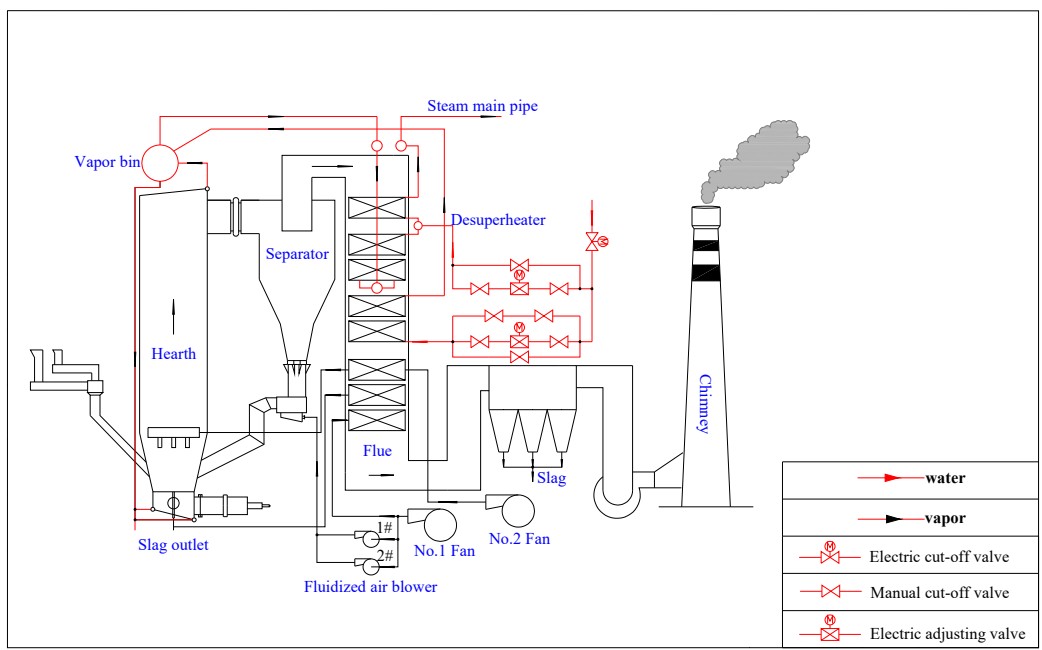

**Figure 4.** Structure of the circulating fluidized bed boiler.

Taking 1460 pieces of data produced by the company from 1 January 2018 to 31 December 2021 as an example, this paper analyzes the multi-dimensional influencing factors of its carbon emissions in order to establish a generalized regression prediction model of carbon emissions from coal-fired power plants in North China. The example of variable names is shown in Table 1 and the example of data is shown in Table 2. (Because the $CO_2$ emissions $X_{12}$ needs to be calculated, the data in Table 2 do not include the $CO_2$ emissions $X_{12}$.)

**Table 1.** Example of carbon emissions variable names of coal-fired power plants.

| Variable | Variable Name |
| --- | --- |
| $X_1$ | Acid consumption (t) |
| $X_2$ | Alkali consumption (t) |
| $X_3$ | Boiler feed water quantity (t) |
| $X_4$ | Incoming raw water quantity (t) |
| $X_5$ | Demineralized water quantity (t) |
| $X_6$ | Elemental carbon content (tC) |
| $X_7$ | Coal consumption (t) |
| $X_8$ | Received base ash (%) |
| $X_9$ | Average low calorific value (kJ/kg) |
| $X_{10}$ | Fly ash carbon content (tC/t) |

**Table 1.** *Cont.*

| Variable | Variable Name |
|---|---|
| $X_{11}$ | Slag carbon content (tC/t) |
| $X_{12}$ | $CO_2$ emissions (t) |

**Table 2.** Example of carbon emissions data of coal-fired power plants.

| X1 | X2 | X3 | X4 | X5 | X6 | X7 | X8 | X9 | X10 | X11 |
|---|---|---|---|---|---|---|---|---|---|---|
| 15.14 | 12.95 | 46,685 | 40,465 | 26,224 | 0.45 | 6109 | 23.75 | 21,272 | 3.21 | 2.15 |
| 32.22 | 8.89 | 47,051 | 40,147 | 26,074 | 0.51 | 5946 | 20.65 | 22,904 | 2.54 | 2.33 |
| 26.87 | 10.41 | 46,637 | 38,953 | 25,906 | 0.48 | 6007 | 21.43 | 22,579 | 2.11 | 1.77 |
| 21.66 | 10.97 | 46,212 | 40,071 | 25,248 | 0.56 | 5517 | 19.11 | 23,413 | 1.45 | 3.11 |
| 17.17 | 9.53 | 46,695 | 41,572 | 27,887 | 0.53 | 5591 | 18.03 | 23,712 | 2.01 | 2.65 |

### 4.2. Analysis of Carbon Emissions Sources

The direct carbon emissions source components of the $CO_2$ emissions boundary of coal-fired power plants include fossil fuel combustion, coal desulfurization, and purchased power. In addition, the $CO_2$ emissions boundary also contains some indirect elements, such as the boiler water purification process, carbon containing residues, etc., as shown in Table 3.

**Table 3.** Analysis of carbon emissions sources of coal-fired power plants.

| Direct Source of Carbon Emissions | | |
|---|---|---|
| **Production Processes** | **Emissions Source** | **Major Greenhouse Gases** |
| Fossil fuel burning | Fossil fuel combustion (burning raw coal, burning oil, burning asphalt, burning blue charcoal) | $CO_2$, $N_2O$, $CH_4$ |
| Coal desulfurization | Consume limestone | $CO_2$ |
| Outsourced electricity | No external purchase of electricity | NAN |
| Carbon residue | Carbon-containing fly ash, carbon-containing slag | $CO_2$, $N_2O$, $CH_4$ |
| **Indirect carbon emissions sources** | | |
| **Production processes** | **Emissions source** | **Major greenhouse gases** |
| Boiler feed water | Acid consumption, alkali consumption, demineralized water consumption, Raw water consumption, boiler feed water | $CO_2$ |

The direct carbon emissions source can directly affect total carbon emissions without a complex chemical conduction process. The indirect carbon emissions source will not directly affect total carbon emissions, but will affect the direct carbon emissions source through chemical conduction, and then affect total carbon emissions. Therefore, in combination with the carbon emissions data of coal-fired power plants in Section 3.1, a sangi diagram is drawn to describe the data flow of carbon emissions from direct carbon emissions sources in each link, as shown in Figure 5. Table 4 provides an example of variable names.

**Table 4.** Examples of variable names of direct carbon emissions sources.

| Variable | Variable Name |
|---|---|
| $Y_1$ | Carbon emissions |
| $Y_2$ | Fossil fuel carbon emissions |
| $Y_3$ | Desulfurization carbon emissions |
| $Y_4$ | Coal burning carbon emissions |

**Table 4.** *Cont.*

| Variable | Variable Name |
|---|---|
| $Y_5$ | Fuel carbon emissions |
| $Y_6$ | Coal-fired carbon emissions from the first power plant |
| $Y_7$ | Coal-fired carbon emissions from the second power plant |
| $Y_8$ | Carbon emissions from coal burning in grinding plants |
| $Y_9$ | Carbon emissions from fuel oil of the first power plant |
| $Y_{10}$ | Carbon emissions from fuel oil of the second power plant |

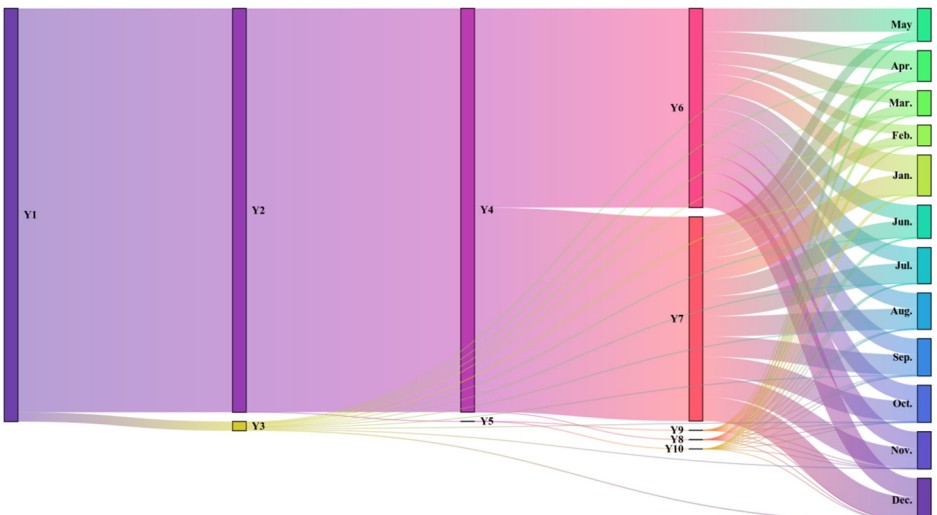

**Figure 5.** Data flow of carbon emissions from direct carbon emissions sources in various links.

## 5. Analysis of Influencing Factors

The design coal of the boiler is generally inferior coal with low carbon content. The amount of $CO_2$ produced by the combustion of unit mass coal depends on the carbon content of coal and the oxidation degree of carbon when burning in the boiler. Fossil fuels of boiler units include coal and various types of fuel oil; however, considering that fuel oil consumption is very small relative to coal consumption and can be ignored (see Figure 4), this paper regards the carbon emissions generated by coal combustion of units as the carbon emissions generated by fossil fuel consumption, and analyzes the significance of relevant influencing factors, in order to screen out the indicators that have a significant impact on the carbon emissions of coal-fired power plants. In the process of data analysis, the influencing factors related to the boiler water circulation system show a high correlation with the carbon emissions results of coal-fired power plants. Therefore, the influencing factors related to the boiler water circulation system are introduced as auxiliary variables to establish the regression prediction model.

### 5.1. Calculation Formulas for Carbon Emissions of Coal-Fired Power Plants

The $CO_2$ emissions of power generation enterprises mainly include three parts: emissions from fossil fuel combustion, $E_{Burn}$; emissions from the desulfurization process, $E_{Des}$; net emissions from electricity purchased and used, $E_{Pur}$.

(1) Emissions from fossil fuel combustion, $E_{Burn}$

$$E_{Burn} = \sum_i (AD_i \times EF_i) \tag{11}$$

where $AD_i$—Activity level of the $i$th fossil fuel (calorific value, TJ). Taking coal-fired power plants as an example, the emissions from fossil fuel combustion of enterprises include coal start-up and stable fuel oil (gas), mobile source oil, etc.

$EF_i$—Emission factor of the ith fossil fuel (tCO$_2$/TJ).

$$AD_i = FC_i \times NCV_i \times 10^{-6} \qquad (12)$$

$FC_i$—Consumption of the ith fossil fuels (t or 10 Nm$^3$);
$NCV_i$—Average low heating value of the ith fossil fuel (kJ/kg or kJ/Nm$^3$).

$$EF_i = CC_i \times OF_i \times 44/12 \qquad (13)$$

$CC_i$—Carbon per unit calorific value of the ith fossil fuels (tC/TJ);
$OF_i$—Carbon oxidation rate (%) of the ith fossil fuel.
For coal burning, the carbon conversion rate $OF_{Coal}$ (%) is:

$$OF_{Coal} = 1 - (G_{Dre} \times C_{Dre} + G_{Ash} \times C_{Ash})/(FC_{Coal} \times NCV_{Coal} \times CC_{Coal}) \qquad (14)$$

where $G_{Dre}$—Slag production (t) calculated at the same time period as coal consumption;
$C_{Dre}$—Average carbon content of slag (%);
$G_{Ash}$—Production of fly ash in the same time period as coal consumption (t);
$C_{Ash}$—Average carbon content of fly ash (%);
$FC_{Coal}$—Consumption of coal (t);
$G_{Ash}$—Average low heating value of coal (kJ/kg);
$CC_{Coal}$—Carbon content per unit calorific value of coal burning (tC/TJ).
The carbon oxidation rate of fuel oil and gas is recommended.
(2) Emissions from the desulfurization process, $E_{Des}$
If carbonate is used as desulfurizer in the desulfurization process, carbon dioxide emissions will be generated:

$$E_{Des} = \sum\nolimits_k (CAL_k \times EF_k) \qquad (15)$$

where $CAL_k$—Carbonate consumption in the kth desulfurizer (ton), and the default value of carbonate content in desulfurizer is 90%.
$EF_k$—Emission factor of carbonate in the kth desulfurizer (tCO$_2$/t)

$$EF_k = EF_{kt} \times TR \qquad (16)$$

where $EF_{kt}$—Emission factor of the desulfurization process during complete conversion (tCO$_2$/t).
$TR$—Conversion rate (%), generally 100%.
(3) Net emissions from electricity purchased and used, $E_{Pur}$
$E_{Pur}$—Carbon dioxide emissions from net purchases of electricity (t)

$$E_{Ele} = AD_{Ele} \times EF_{Ele} \qquad (17)$$

where $AD_{Ele}$—Net electricity purchased by the enterprise (MWh);
$EF_{Ele}$—Annual average power supply emission factor of the regional power grid (tCO$_2$/MWh).
The annual average power supply emission factor of the regional power grid varies greatly from province to province. In Beijing, according to 0.604 tCO$_2$/MWh, coal-fired power plants with more than two sets of units generally do not need to purchase power because they can be used as back-up for each other.

### 5.2. Analysis of Influencing Factors of Direct Carbon Emissions

It can be seen from Table 2 that the direct influencing factors of carbon emissions of coal-fired power plants include the carbon content of slag, carbon content of fly ash, elemental carbon content, average low calorific value, coal consumption, and received base ash. The main purpose of this section is to screen the influencing factors that have a

significant impact on the carbon emissions results from the above six influencing factors, which will be used as part of the input set of the later coal-fired power plant carbon emissions regression prediction model.

5.2.1. Establishment of Simulation Model of Direct Influencing Factors of Carbon Emissions from Coal-Fired Power Plants

According to Section 5.1, there are multiple feedback loops in the coal-fired carbon emissions system of the thermal power plant. The causality diagram of the coal-fired carbon emissions system of the thermal power plant established according to the causality between the feedback loop and each variable is shown in Figures 6–9.

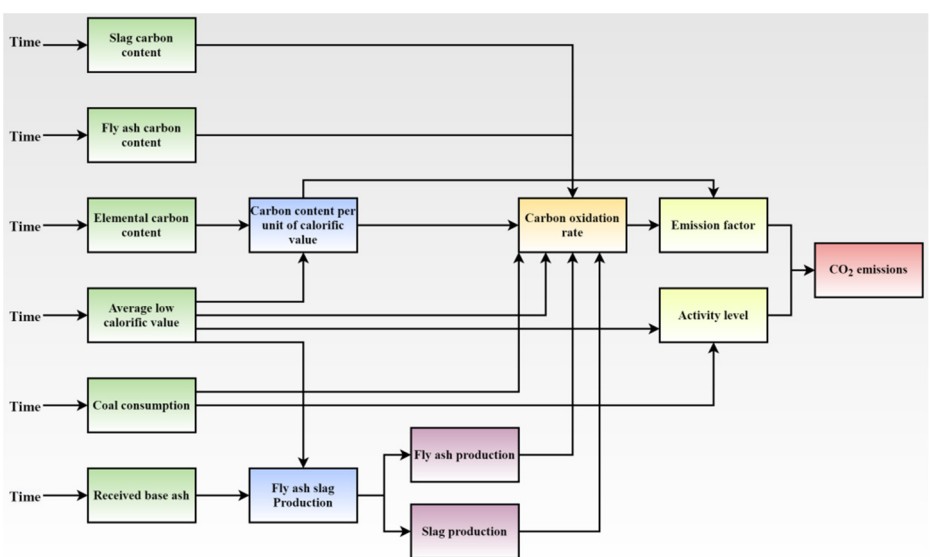

**Figure 6.** Causes tree of coal-fired carbon emissions system in thermal power plants.

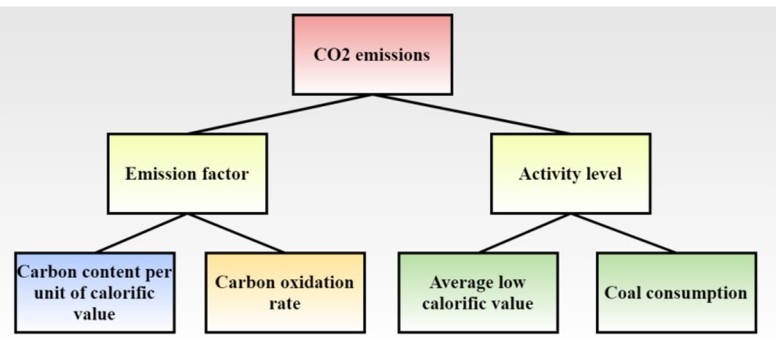

**Figure 7.** Uses tree of coal-fired carbon emissions in thermal power plants.

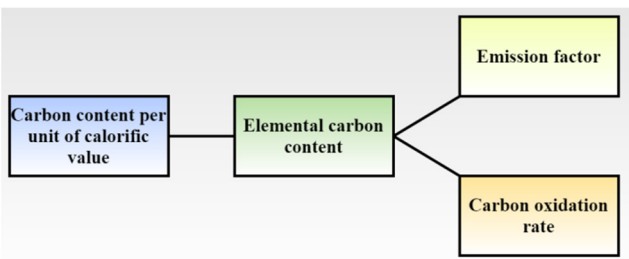

**Figure 8.** Uses tree of element carbon content.

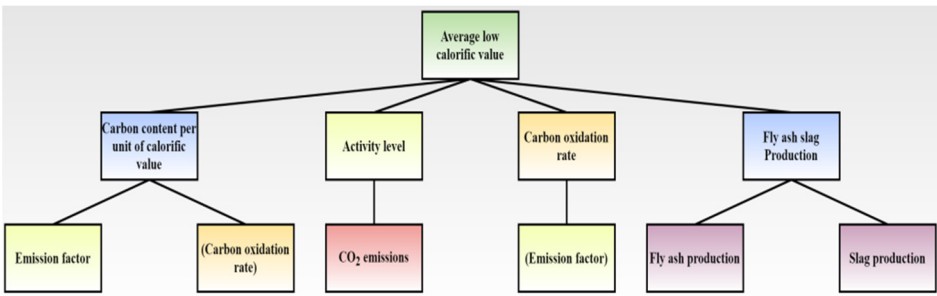

**Figure 9.** Uses tree of average low calorific value.

Using Vensim software and existing data, the carbon emissions results of coal-fired power plants are simulated and calculated. The time step is one day and the completion time of simulation is one year. The calculation formula of coal-fired carbon emissions is shown in Section 5.1, with the relevant data examples shown in Table 1. The simulation results of the key parameters carbon content per unit calorific value, carbon oxidation rate, and coal-fired carbon emissions are as follows.

(1) Carbon content per unit calorific value

Through the simulation of the carbon emissions system of the coal-fired power plant, the simulation results of the carbon content per unit calorific value of the power plant in 2021 are obtained, as shown in Figure 10.

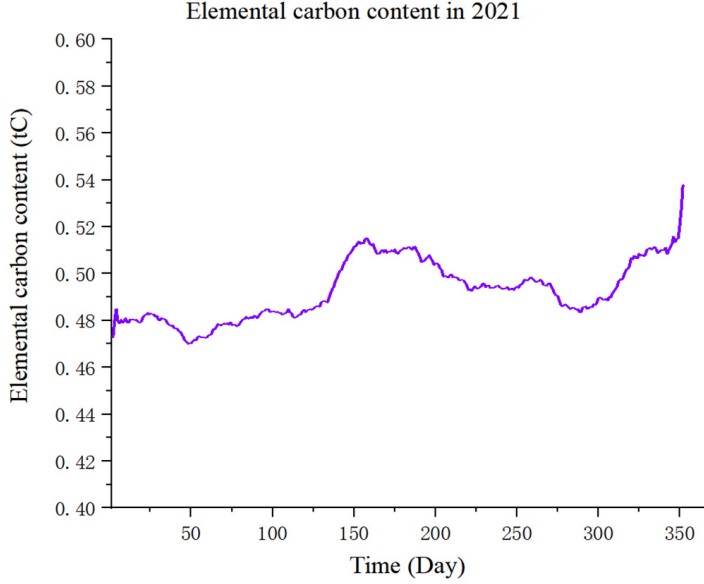

**Figure 10.** Simulation results of carbon content per unit calorific value in 2021.

(2) Carbon oxidation rate

Through the simulation of the carbon emissions system of the coal-fired power plant, the simulation data of the carbon oxidation rate of the power plant in 2021 are obtained, as shown in Figure 11.

(3) Carbon emissions from coal-fired thermal power plants

Through the simulation of the carbon emissions system of the coal-fired power plant, the simulation results of carbon emissions from coal combustion of the power plant in 2021 are obtained, as shown in Figure 12.

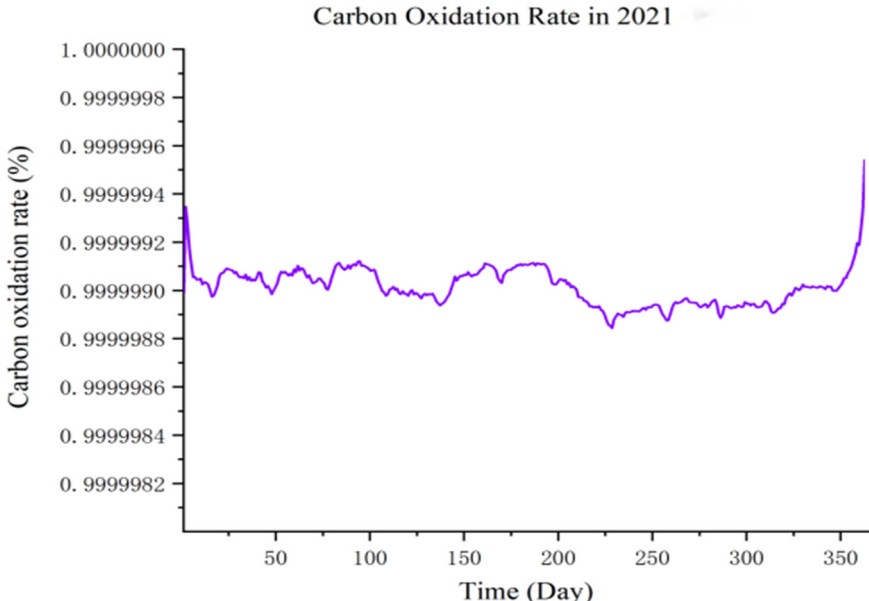

**Figure 11.** Simulation results of carbon oxidation rate in 2021.

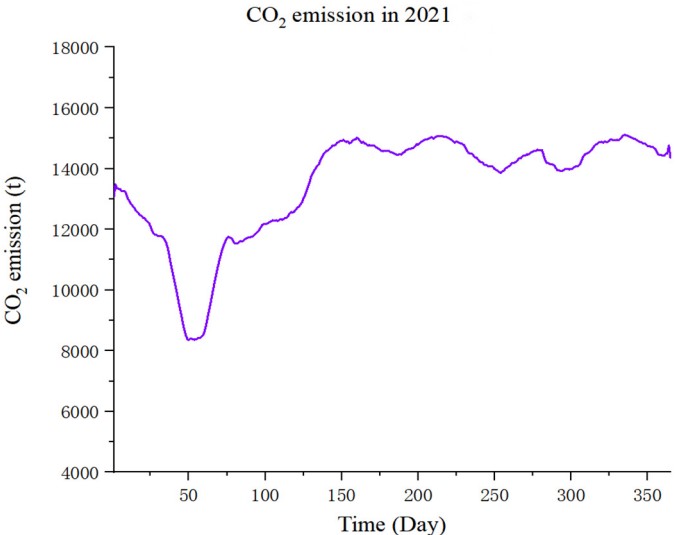

**Figure 12.** Simulation results of coal-fired carbon emissions in thermal power plants in 2021.

### 5.2.2. Significance Analysis of Direct Influencing Factors of Carbon Emissions from Coal-fired Power Plants

From the schematic diagram of the influence path of system parameters and the simulation results of the model, it can be seen that the six parameters of elemental carbon content, coal consumption, received base ash, average low calorific value, carbon content of fly ash, and carbon content of slag are the most original factors affecting the carbon emissions in the carbon emissions system of coal-fired power plants, and the degree of influence is different. In this paper, the influence degree of each factor on the carbon emissions of coal-fired power plants is measured by setting a certain degree (+5%) of the initial deviation. The simulation results of the model are as follows.

(1)　Received base ash (+5%): the effect is not significant

It can be seen from Figure 13 that an increase of +5% of received base ash has a slight impact on the carbon oxidation rate, but has a weak impact on total carbon emissions.

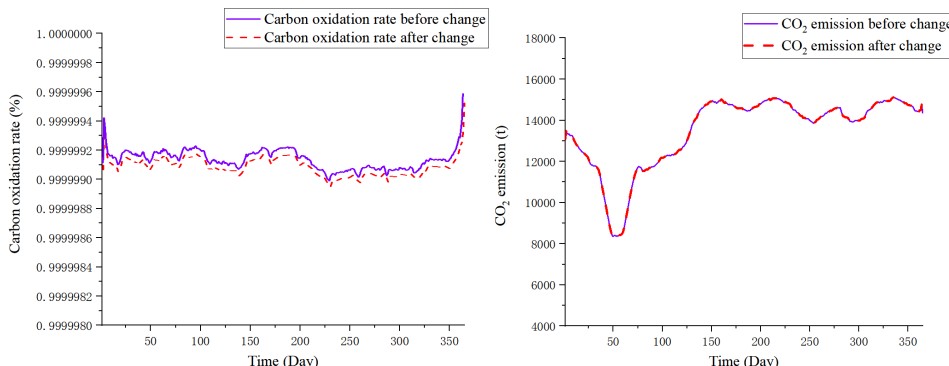

**Figure 13.** Comparison of the influence degree of received base ash.

(2)    Element carbon content (+5%): significant impact

It can be seen from Figure 14 that the increase of element carbon content +5% has a great impact on the emission factor, which indirectly affects the carbon emissions of coal combustion, with the degree of impact relatively significant.

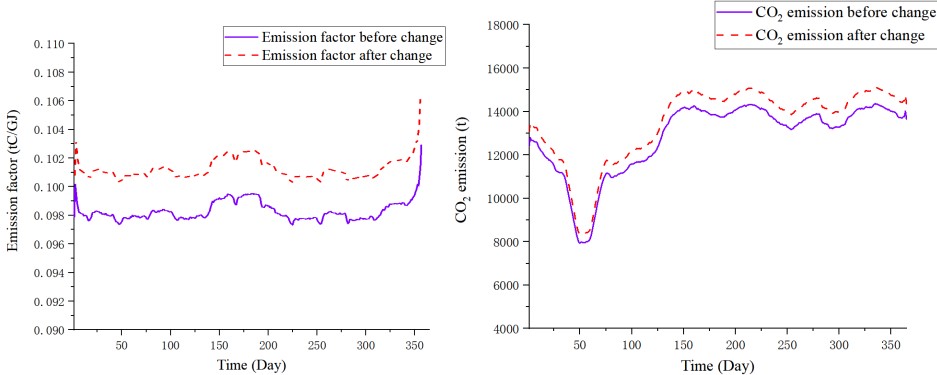

**Figure 14.** Comparison of the influence degree of elemental carbon content.

(3)    Average low calorific value (+5%): the effect is not significant

It can be seen from Figure 15 that the change of the average low value (+5%) increases the activity level and reduces the value of the emission factor. Therefore, there is a certain offset between them, resulting in the final impact on the total carbon emissions not being significant.

(4)    Coal consumption (+5%): significant impact

It can be seen from Figure 16 that the change of coal consumption directly affects the change of activity level, thus significantly affecting the carbon emissions of coal combustion.

(5)    Carbon content of fly ash (+5%): the effect is not significant

It can be seen from Figure 17 that the carbon content of fly ash has a slight effect on the carbon oxidation rate, but the effect on total carbon emissions is not significant.

(6)    Carbon content of ash (+5%): the effect is not significant

It can be seen from Figure 18 that the carbon content of fly ash has a slight effect on the carbon oxidation rate, but the effect on total carbon emissions is not significant.

In this section, the direct factors affecting the carbon emissions of coal-fired thermal power plants are analyzed. The simulation calculation and image results show that the carbon oxidation rate will be affected to a certain extent when the base ash content, carbon content of fly ash, and carbon content of ash slag are increased. However, since the carbon oxidation rate value is generally stable, the final impact on the carbon emissions of coal-fired power plants is not significant. When the average low calorific value increases, on

the one hand, it will lead to an increase of activity level; on the other hand, it will reduce the emission factor, so the impact on coal-fired carbon emissions under the comprehensive effect is not significant. Under the same adjustment range, the increase of elemental carbon content and coal consumption will significantly affect coal-fired carbon emissions by affecting emission factors and activity levels, respectively.

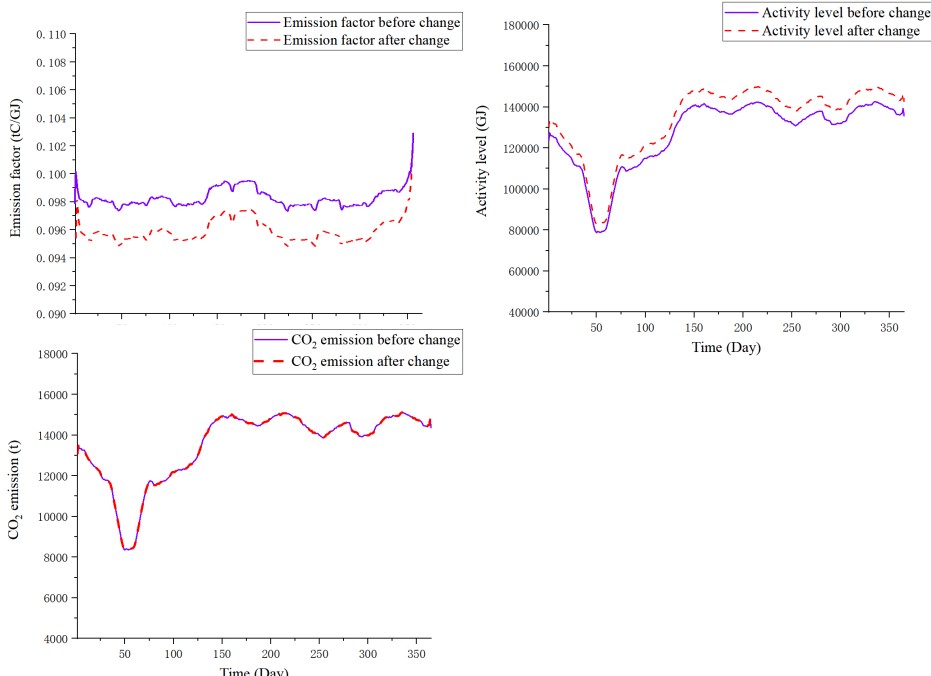

**Figure 15.** Comparison of the influence degree of the average low calorific value.

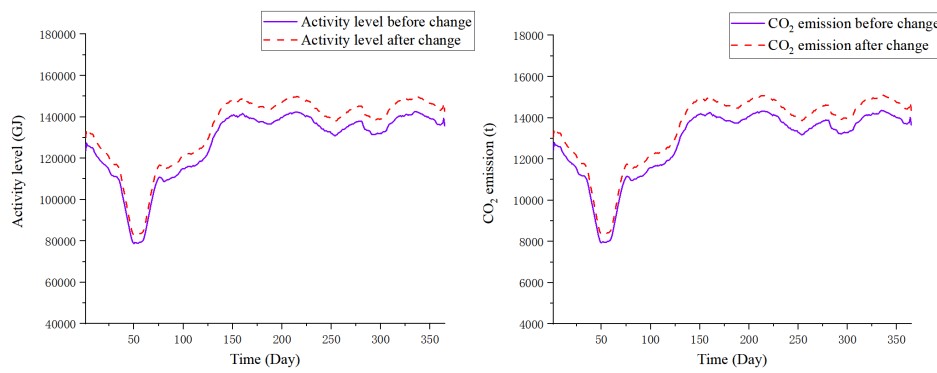

**Figure 16.** Comparison of the influence degree of coal consumption.

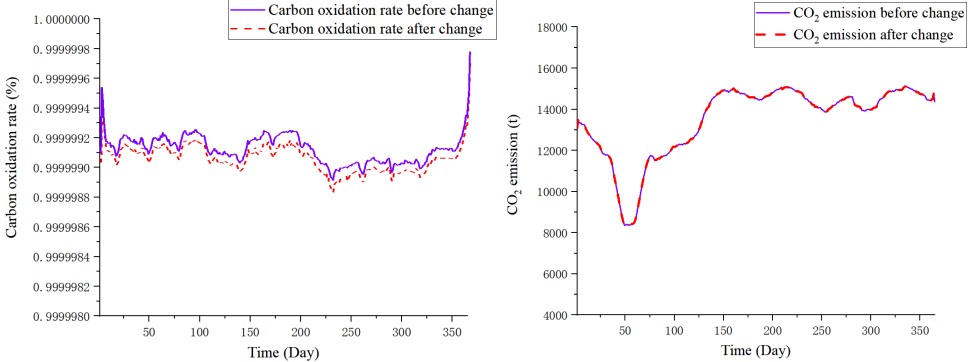

**Figure 17.** Comparison of the influence degree of carbon content in fly ash.

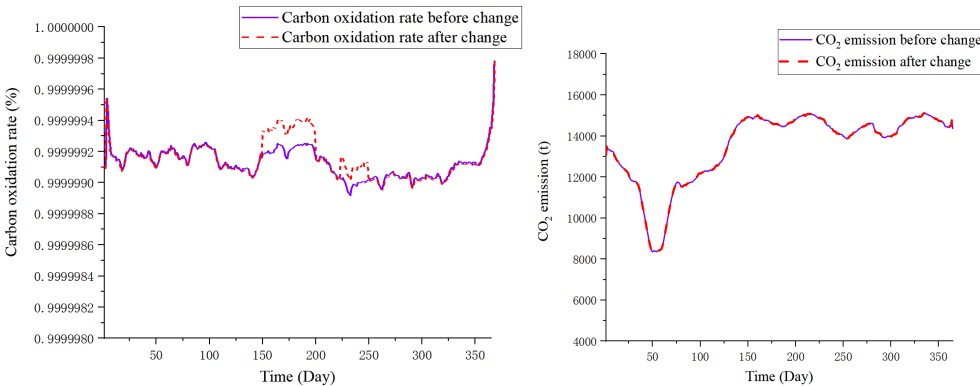

**Figure 18.** Comparison of the influence degree of carbon content in ash and slag.

### 5.3. Analysis of Influencing Factors of Carbon Emissions of Coal-Fired Power Plants Considering Boiler Feed Water System

The XG-Boost algorithm can extract features from many relevant historical data, and then realize the characteristic importance analysis of various influencing factors on carbon emissions of coal-fired power plants. This section collects the acid consumption, alkali consumption, boiler water supply, raw water consumption, and demineralized water consumption during boiler operation as auxiliary influencing factors, and then combines the direct influencing factors of carbon emissions of coal-fired power plants to construct a complete input set of a regression prediction model.

#### 5.3.1. Correlation Analysis of Influencing Factors of Boiler Feed Water System

In order to explore the correlation between acid consumption, alkali consumption, boiler water supply, water consumption, and desalted water consumption with carbon emissions of coal-fired power plants, correlation analysis was performed on historical data.

It can be seen from Figure 19 and Table 5 that the correlation between acid consumption, alkali consumption, boiler water supply, raw water consumption, and desalted water consumption with carbon emissions of coal-fired power plants is 0.42, 0.41, 0.96, 0.75, and 0.57, respectively, and the significance level is 0.01. Therefore, the above impact factors have a strong correlation with carbon emissions of coal-fired power plants.

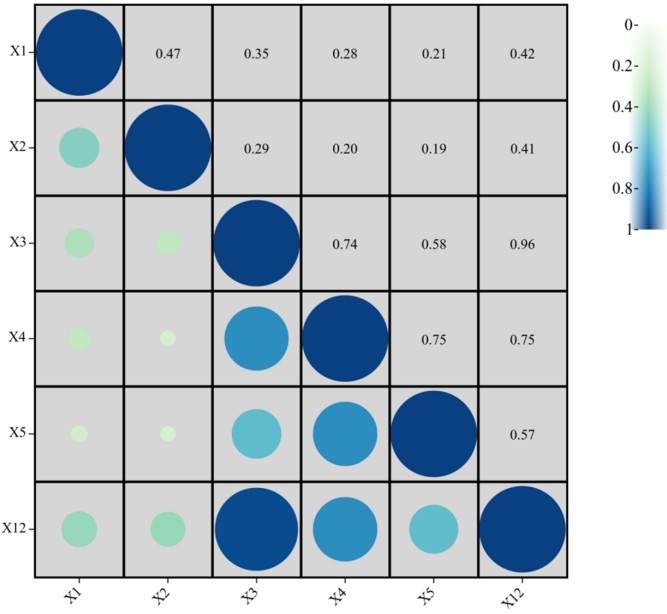

**Figure 19.** Heat map of the correlation analysis between the influencing factors of the boiler feed water system and carbon emissions.

**Table 5.** Correlation analysis and significance test of influencing factors of boiler feed water system and carbon emissions.

| Impact Factor | Correlation | Significance Test |
|---|---|---|
| Acid consumption ($X_1$) | 0.42 | 0.01 |
| Alkali consumption ($X_2$) | 0.41 | 0.01 |
| Boiler feed water quantity ($X_3$) | 0.96 | 0.01 |
| Incoming raw water quantity ($X_4$) | 0.75 | 0.01 |
| Demineralized water quantity ($X_5$) | 0.57 | 0.01 |
| $CO_2$ emissions ($X_{12}$) | 1 | 0 |

5.3.2. Characteristic Importance Analysis of Influencing Factors of Boiler Feed Water System Based on XG-Boost Algorithm

This section introduces the elemental carbon content and coal consumption screened in 5.2 to comprehensively analyze the characteristic importance of relevant influencing factors of the boiler feed water system in the carbon emissions process of coal-fired power plants. The XG-Boost algorithm is used to clearly calculate each influencing factor in the carbon emissions dataset of coal-fired power plants, in order to sort and compare the auxiliary influencing factors and direct influencing factors. Among them, the auxiliary influencing factors include $X_1$ acid consumption, $X_2$ alkali consumption, $X_3$ boiler water supply, $X_4$ raw water consumption, and $X_5$ demineralized water consumption, while the direct influencing factors include $X_6$ element carbon content and $X_7$ coal consumption. The characteristic importance of a single decision tree is calculated by increasing the number of performance indicators at the split point of each influencing factor, weighted by the observation times of the node. Finally, the characteristic importance is averaged among all the decision trees in the model, in order to obtain the characteristic importance of the influencing factors of the boiler feed water system on the carbon emissions of the coal-fired power plant. The results of the characteristic importance are shown in Figure 20 and Table 6.

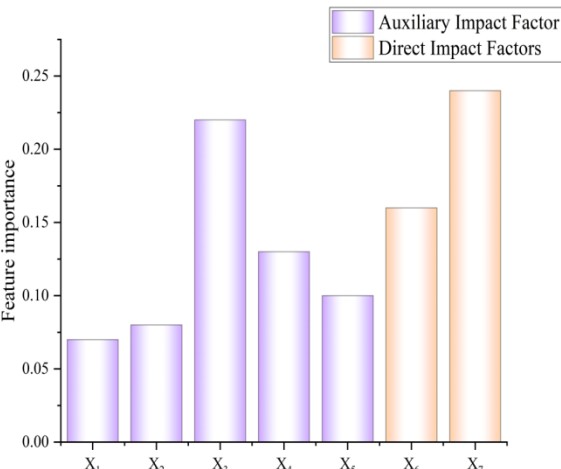

**Figure 20.** Characteristic importance results of the influencing factors of the boiler feed water system on carbon emissions of coal-fired power plants.

It can be seen from Figure 20 and Table 6 that the characteristic importance of auxiliary influencing factors is 0.6 and that of direct influencing factors is 0.4. The auxiliary influencing factor of the boiler feed water system introduced in this paper has a high characteristic importance to the carbon emissions of coal-fired power plants, while there is no extreme unbalanced characteristic importance result on the whole.

In summary, this paper selects acid consumption, alkali consumption, boiler water supply, raw water consumption, demineralized water consumption, elemental carbon content, and coal consumption as the input set of the carbon emissions prediction model of

coal-fired power plants in the following text, in order to establish a more comprehensive multi-dimensional carbon emissions regression prediction model of coal-fired power plants.

**Table 6.** Numerical results of the characteristic importance degree of the influencing factors of the boiler feed water system on the carbon emissions of coal-fired power plants.

| First-Level Impact Factor | Second-Level Impact Factor | Feature Importance | Feature Importance |
|---|---|---|---|
| Auxiliary Impact Factor | Acid consumption ($X_1$) | 0.07 | 0.6 |
| | Alkali consumption ($X_2$) | 0.08 | |
| | Boiler feed water quantity ($X_3$) | 0.22 | |
| | Incoming raw water quantity ($X_4$) | 0.13 | |
| | Demineralized water quantity ($X_5$) | 0.10 | |
| Direct impact factor | Elemental carbon content ($X_6$) | 0.16 | 0.4 |
| | Coal consumption ($X_7$) | 0.24 | |

## 6. Establishment and Empirical Analysis of SSA-LSTM Regression Prediction Model

Firstly, in order to verify the superiority of the regression prediction model established by the LSTM algorithm optimized based on the sparrow algorithm (SSA), this section establishes the regression prediction model by introducing the BP neural network algorithm and the LSTM algorithm, compares it with the optimized LSTM regression prediction model, and then inputs the above screened $X_1$ acid consumption, $X_2$ alkali consumption, $X_3$ boiler water supply, $X_4$ water consumption, $X_5$ demineralized water consumption, $X_6$ element carbon content, and $X_7$ coal consumption into the comparison model, to verify the effect of the model. Secondly, in order to verify the superiority of the regression prediction model established by introducing the influencing factors of the boiler feed water system, this paper uses six direct influencing factors to establish a BP neural network regression prediction model, LSTM regression prediction model, and SSA-LSTM regression prediction model to verify the superiority of the index.

### 6.1. Model Parameter Setting and Establishment

The structure of the BP neural network includes one input layer, two hidden layers, and one output layer. The dimension of the input layer is three and the number of neurons in the middle two hidden layers is 100. Firstly, the dimension is increased through hidden layer 1 with 100 neurons in order to obtain rich shallow features, then the dimension is reduced through hidden layer 2 composed of 100 neurons to obtain deeper features and reduce the dimension of output. The dimension of output neurons is 1. The BP optimization algorithm will use the random gradient descent optimization algorithm to optimize the parameters of the whole network, constantly reduce the gap between the output of the network and y value, and adjust the weight. The super parameters in the training process are as follows: set the learning rate of the network to 0.001, the number of iterations of training to 50, the batch size to 256, and use GPU for parallel acceleration. As a comparison, this paper constructs the same two-layer LSTM network structure, with the input and output dimensions also consistent with the BP neural network, and the training, verification, and test data also consistent. Super parameters such as the learning rate, the number of training iterations, batch size, the number of nodes in the first hidden layer, the number of nodes in the second hidden layer, and the running hardware environment (GPU Parallel Computing) are the same as BP, and the difference is that the hidden layer is replaced by the LSTM module. Finally, the sparrow algorithm (SSA) is used to find the optimal LSTM super parameters, in order to improve the accuracy of the whole prediction. The optimized super parameters include the learning rate, number of training iterations, number of nodes in the two LSTM hidden layers, and batch size. By taking the predicted value of $CO_2$ emissions and the mean square deviation of the real $CO_2$ emissions as the fitness function, this function is optimized in order to find a set of super parameters that can minimize the error of the whole network. The optimal learning rate is 0.009659, the

optimal training time is 50, the optimal first LSTM hidden layer is 73 nodes, the optimal second LSTM hidden layer is 20 nodes, and the batch size is 256. The model of GPU is Nvidia 2070 single card training, the depth learning box used is pytorch1.5, and the input data are the production logs of a large thermal power plant from 1 January 2018 to 31 December 2021, with a total of 1460 entries.

### 6.2. Establishment of Regression Model Based on SSA-LSTM Algorithm

The previous article analyzed the historical data of carbon emissions from coal-fired power plants through a series of methods, and established the following three datasets:

Real dataset A: $X_{12}$ $CO_2$ emissions.

Dataset B of influencing factors introduced into the boiler feed water system: $X_1$ acid consumption, $X_2$ alkali consumption, $X_3$ boiler feed water, $X_4$ raw water consumption, $X_5$ demineralized water consumption, $X_6$ elemental carbon content, $X_7$ coal consumption, $X_{12}$ $CO_2$ emission.

Dataset C with only six original direct influencers: $X_6$ element carbon content, $X_7$ coal consumption, $X_8$ received base ash, $X_9$ average low calorific value, $X_{10}$ fly ash carbon content, $X_{11}$ slag carbon content, $X_{12}$ $CO_2$ emissions.

Therefore, this section selects the above three datasets to establish the SSA-LSTM regression prediction model, with the specific process as follows.

LSTM regression model steps optimized by the sparrow algorithm (SSA):

Step 1: Divide the dataset into a training set and a test set, in which the training set accounts for 95% of the total dataset and is 1387 samples; the test set accounts for 5% of the total dataset, which is 73 samples;

Step 2: Use the SSA algorithm to optimize the super parameters of the LSTM regression prediction model;

Step 3: Initialize the population and the number of iterations, and initialize the comparison of predators and accessors;

Step 4: Calculate the fitness value and sort it;

Step 5: Use Formula (8) to update the predator position;

Step 6: Use Formula (9) to update the position of participants;

Step 7: Use Formula (10) to update the position of the vigilant;

Step 8: Calculate the fitness value and update the sparrow position;

Step 9: Judge whether the stop conditions are met. If so, exit and output the results. Otherwise, repeat steps 4–8;

Step 10: Build the LSTM regression model of carbon emissions of coal-fired power plants by combining the training samples and the obtained optimal parameter combination;

Step 11: Input the test set into the regression model and output the fitting curve of carbon emissions of coal-fired power plants.

### 6.3. Model Effect Display and Comparison

Taking dataset B of the influencing factors of the boiler feed water system as the input dataset, the regression prediction results of the three models are shown in Figure 21, in which the abscissa is time and the ordinate is $CO_2$ emissions. SSA-LSTM has a stronger fitting ability to extreme points, so it has achieved the optimal fitting effect. Table 7 shows the specific algorithm performance results, where MAE represents the average absolute error, RMSE represents the root mean square error, MAPE represents the average absolute percentage error, and $R^2$ represents the goodness of fit. The smaller the MAE, the better the model effect. The smaller the RMSE, the better the model effect. The smaller the MAPE, the better the model effect. The larger the $R^2$ value, the better the fitting ability of the model.

Taking the dataset of the influencing factors of the boiler feed water system and the dataset containing only six direct influencing factors as the input datasets of the three algorithms, the regression prediction effect is shown in Figures 22–24. In Figures 22–24, the X axis is time, the Y axis represents the type of dataset (where A represents the true value, B represents the dataset of the influencing factors of the boiler feed water system, and C

represents the dataset containing only six original direct influencing factors), and the Z axis is $CO_2$ emissions. The specific algorithm performance results are shown in Table 8.

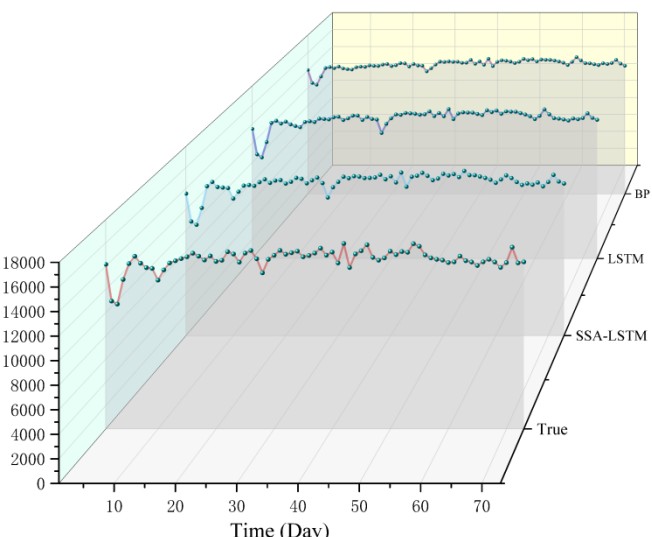

**Figure 21.** Comparison of model fitting effects with the introduction of influencing factors of the boiler feed water system.

**Table 7.** Numerical results of model performance with influencing factors of the boiler feed water system introduced.

|  | MAE | RMSE | MAPE | $R^2$ |
|---|---|---|---|---|
| BP | 634.83 | 726.47 | 0.054 | 0.623 |
| LSTM | 312.33 | 380.03 | 0.026 | 0.809 |
| SSA-LSTM | 289.14 | 357.15 | 0.021 | 0.842 |

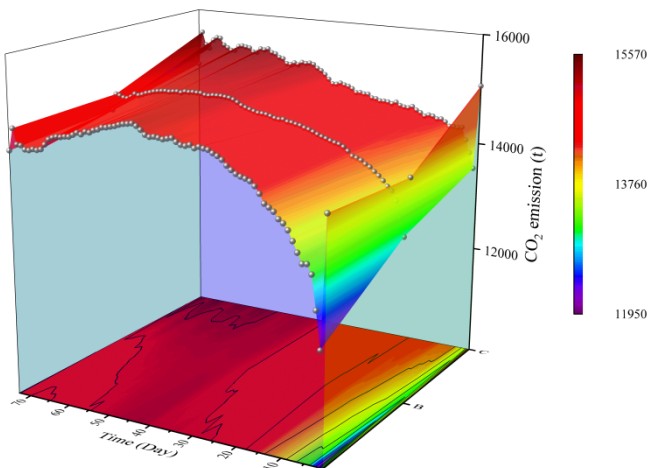

**Figure 22.** The fitting effect of the BP neural network algorithm on different datasets.

The contour lines at the bottom of Figures 22–24 can intuitively reflect the differences between dataset B and dataset C and real dataset A. Through three groups of comparative experiments, it is obvious that the predicted value of the regression prediction model established by using dataset C containing only six direct influencing factors is always not within the same contour line with the real value, and there is a significant difference, while the predicted value of the regression prediction model established by using the dataset introducing the influencing factors of the boiler feed water system is always within the same contour line, and the difference is small. It can be seen from Table 8 that the

performance of the model using dataset B as the input set is better than that using dataset C, indicating that the introduction of influencing factors of the boiler feed water system as the prediction of carbon emissions from coal-fired power plants can improve the prediction effect. Combined with the SSA-LSTM algorithm, the effect of the model can be further improved. Moreover, the contour of the regression prediction model established by the SSA-LSTM algorithm is smoother, which verifies that the robustness of the SSA-LSTM algorithm is better than the other two algorithms.

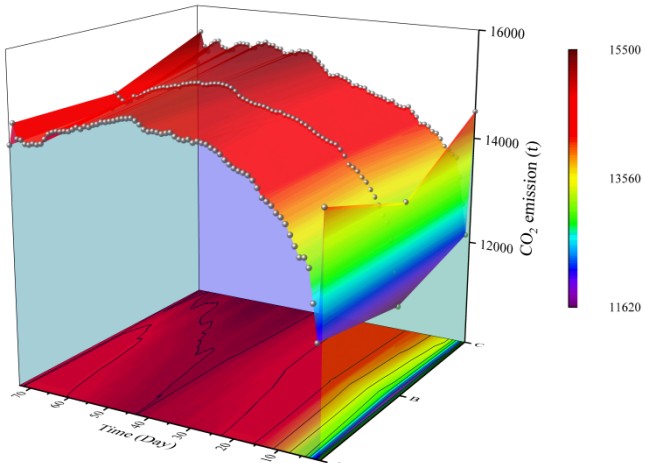

**Figure 23.** The fitting effect of the LSTM algorithm on different datasets.

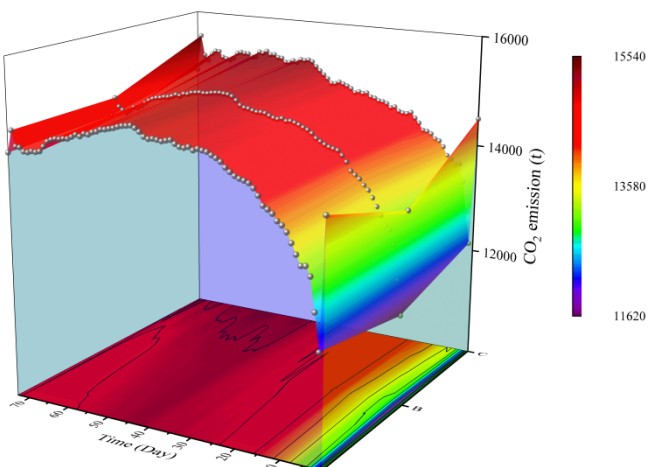

**Figure 24.** The fitting effect of the SSA-LSTM algorithm on different datasets.

**Table 8.** Numerical results of the performance of each model under different datasets.

| Dataset | Algorithm | MAE | RMSE | MAPE | $R^2$ |
|---------|-----------|-----|------|------|-------|
|           | BP       | 634.83 | 726.47 | 0.054 | 0.623 |
| Dataset B | LSTM     | 312.33 | 380.03 | 0.026 | 0.809 |
|           | SSA-LSTM | 289.14 | 357.15 | 0.021 | 0.842 |
|           | BP       | 754.45 | 801.92 | 0.071 | 0.589 |
| Dataset C | LSTM     | 334.22 | 401.57 | 0.032 | 0.786 |
|           | SSA-LSTM | 312.17 | 377.24 | 0.028 | 0.815 |

## 7. Conclusions

This paper takes the carbon emissions of a cogeneration enterprise in North China as the research object, and using its production log from 2019 to 2021 as the data support, establishes a regression prediction model for carbon emissions of coal-fired power plants and, on this basis, provides some emissions reduction policies.

*7.1. Carbon Emissions Regression Prediction Model Results*

Firstly, according to the relevant accounting formula of carbon emissions of coal-fired power plants and the influencing factors of direct carbon emissions, a carbon emissions simulation system of coal-fired power plants based on system dynamics is built using Vensim software, and the influence ways and mutual relations of various parameters in the system are decomposed and analyzed, so that the mutual influence relationship between the influencing factors of direct carbon emissions and their influence on total carbon emissions can be discussed more intuitively. The indexes were screened for establishing the regression prediction model. Secondly, the XG-Boost algorithm is used to calculate the characteristic importance of each influencing factor in the total concentration of carbon emissions indicators of coal-fired power plants, sort and compare the auxiliary carbon emissions influencing factors and direct carbon emissions influencing factors, and propose to introduce the relevant influencing factors of the boiler feed water system to expand the dimensions of the input indicators of the regression prediction model, in order to improve the performance of the regression prediction model. Finally, this paper determines that the acid consumption, alkali consumption, boiler water supply, raw water consumption, demineralized water consumption, elemental carbon content, and coal consumption are the input indicators for establishing the carbon emissions regression prediction model of coal-fired power plants. Secondly, aiming at the blindness of random selection of the regression prediction model parameters, this paper uses the SSA algorithm to iteratively optimize the learning rate and hidden layer of the LSTM algorithm, and then inputs the dataset containing only six direct influencing factors and the dataset introducing the boiler feed water system influencing factors into the BP neural network algorithm, LSTM algorithm, and SSA-LSTM algorithm to establish six regression prediction models. By comparing the performance criteria of six different models, it can be seen that the effect of the model established by using the dataset with the influencing factors of the boiler feed water system has been significantly improved. The model established by using the SSA-LSTM algorithm performs better on the two datasets, which verifies the superiority of the carbon emissions regression prediction model of thermal power plants based on the SSA-LSTM algorithm with the influencing factor of boiler feed water. On the basis of statistical analysis, this paper analyzes the feature importance of influencing factors with the help of the XG-Boost algorithm. Compared with subjective methods such as principal component analysis and analytic hierarchy process, the XG-Boost algorithm can be used to objectively analyze various influencing factors of carbon emissions from coal-fired power plants, making the selection of influencing factors more scientific. The advantage of the SSA-LSTM algorithm is that it can effectively simplify the super-parameter selection process of the LSTM algorithm, effectively solve the global optimization problem, prevent the model from falling into overfitting and local optimization, and achieve a better fitting effect for the regression prediction model of carbon emissions of coal-fired power plants. Therefore, the SSA-LSTM algorithm-based thermal power plant carbon emissions regression prediction model established in this paper, considering the influencing factor of boiler feed water, has better model performance than the traditional model, and can provide a better model reference for the calculation of China's energy consumption carbon emissions, the decomposition of influencing factors, and the analysis of future carbon emissions prediction.

*7.2. Policy Suggestions*

(1) Deepen the research on the decomposition of factors affecting carbon emissions of coal-fired power plants.

The change of carbon emissions of coal-fired power plants is the result of a combination of many factors. The effects of these factors on carbon emissions can be divided into two types: positive driving effect and negative driving effect. That is, the change in some factors will drive an increase of carbon emissions, while changes in others will inhibit the growth of carbon emissions. Only by clarifying these influencing factors and their effects can we

better undertake carbon emissions prediction research of coal-fired power plants on this basis, and then propose scientific and reasonable policy measures for the realization of China's energy conservation and emissions reduction goals.

(2)     Optimization of carbon emission prediction model for coal-fired power plants.

The prediction of carbon emissions from coal-fired power plants can provide a basic understanding and grasp of the evolution trend of carbon emissions in the future. Through big data technology, optimize the carbon emissions regression prediction model of coal-fired power plants, establish a multi-scale model for rapid and accurate estimation of carbon emissions, and combine the actual energy utilization background with reference to the national economic and social development goals, then set possible scenarios in the future and make more accurate measurement of carbon emissions reduction, in order to promote the optimization of China's energy structure.

According to the actual situation, China's power sector can generalize the model established in this paper by increasing or decreasing the influencing factors, adjusting the parameters, and optimizing the algorithm, and then simulate the change trend of China's power sector's carbon emissions in the future by combining the scenario analysis method, analyze according to the prediction results, and formulate targeted emissions reduction plans. In future research, based on the research paradigm proposed in this paper, through more innovative and intelligent computing methods, further optimize the carbon emissions regression prediction model of China's power industry, explore the specific process of carbon peaking and carbon neutralization, and propose corresponding solutions for reference.

**Author Contributions:** Conceptualization, X.W.; data curation, C.Y.; computation, C.Y.; data analysis, W.L. and X.L.; writing—original draft preparation, X.W., C.Y. and W.L.; writing—review and editing, X.W., C.Y. and W.L. All authors have read and agreed to the published version of the manuscript.

**Funding:** This work was supported in part by the National Statistical Science Research Project Key Project (2022LZ31), the General Project of Shandong Provincial Natural Science Foundation (ZR2022MG059), the General Project of Shandong Provincial Natural Science Foundation (ZR2020MF033) and the General project of Science and Technology Plan of Beijing Municipal Commission of Education (KM202010017001).

**Institutional Review Board Statement:** Not applicable.

**Informed Consent Statement:** Not applicable.

**Data Availability Statement:** Initial data on carbon emissions are available upon request from CHP companies. The dataset generated during the current study is available from the corresponding author on reasonable request.

**Conflicts of Interest:** The authors declare no competing interests.

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
