# Peer review of "Research on Carbon Emissions Prediction Model of Thermal Power Plant Based on SSA-LSTM Algorithm with Boiler Feed Water Influencing Factors"

_sustainability, doi:10.3390/su142315988_

Round 1

Reviewer 1 Report

The paper establishes a model to predict carbon emissions from thermal power plants. It is based on an interesting idea and presents a relevant topic. However, it has a few areas for improvement. Below is a list of remarks:

1)     Introduction. The general discussion in the introduction should be limited. There is no need to repeat well-known issues. The introduction should include problem context, literature review and the hypothesis based on the gap analysis of the previously published research. Emphasis should be on a targeted literature review and a justification for the research that your paper is going to cover (i.e. what is the scientific gap?). The introduction should include problem context, literature review and the hypothesis based on the gap analysis of the previously published research.

a.      Brief content. The introduction segment should be enhanced by adding a final paragraph that shortly highlights the key segments of the paper.

2)     Literature review. The authors did not provide a literature review, disregarding previous research in the area. As aforementioned, a thorough literature review should be conducted and the specific science gap outlined.

a.      I would suggest referencing papers that deal with the topic of your paper and exploring the differences in approaches between their models and your. Below is a list of a few papers that explore the area of your research:

                                                    i.     https://doi.org/10.3390/atmos12070811

                                                   ii.     https://doi.org/10.1080/15567249.2022.2037028

                                                  iii.     https://doi.org/10.1016/j.energy.2020.119026

b.      I also believe that there are plenty of researchers outside China that deal with the topic of your paper – why haven’t their research efforts been acknowledged?

3)     Method. Developing a suitable methodology is crucial. The authors should invest more time in ensuring all parameters in the formulae used are stated and elaborated – at present, this is not the case.

4)     Conclusion. The final chapter should be expanded, and practical implications of your research should be elaborated in more detail. It would be meaningful if you would discuss the results by suggesting how they are relevant to further sector development and how your research could be used in practice.

a.      Discussion. This segment should be enhanced with additional outcomes and comparisons with other studies in the field. Are your results aligned with those of other studies in the field? Also, the benefits of the proposed model are not clear enough and should be described more concisely.

Author Response

Response to Reviewer 1 Comments

Point 1: Introduction. The general discussion in the introduction should be limited. There is no need to repeat well-known issues. The introduction should include problem context, literature review and the hypothesis based on the gap analysis of the previously published research. Emphasis should be on a targeted literature review and a justification for the research that your paper is going to cover (i.e. what is the scientific gap?). The introduction should include problem context, literature review and the hypothesis based on the gap analysis of the previously published research.

  1. Brief content. The introduction segment should be enhanced by adding a final paragraph that shortly highlights the key segments of the paper.

Response 1: Thank you. We have rewritten the introduction as you suggested. The relevance of the article, the novelty of the results, the importance of the policy implications, the selection of the sample, the appropriateness of the methodology, the data sources, the contribution to the literature, and the limitations of the study are highlighted.

Point 2:   Literature review. The authors did not provide a literature review, disregarding previous research in the area. As aforementioned, a thorough literature review should be conducted and the specific science gap outlined.

  1. I would suggest referencing papers that deal with the topic of your paper and exploring the differences in approaches between their models and your. Below is a list of a few papers that explore the area of your research:
  2. https://doi.org/10.3390/atmos12070811
  3. https://doi.org/10.1080/15567249.2022.2037028

                                                  iii.     https://doi.org/10.1016/j.energy.2020.119026

  1. I also believe that there are plenty of researchers outside China that deal with the topic of your paper – why haven’t their research efforts been acknowledged?

Response 2: Thank you. We have divided the introduction and literature review into two separate chapters according to your suggestion. By comparing with previous articles, the innovation and limitation of this article are emphasized. And cites the first article you provided, see Ref. [26].

Point 3:  Method. Developing a suitable methodology is crucial. The authors should invest more time in ensuring all parameters in the formulae used are stated and elaborated – at present, this is not the case.

Response 3: Thank you. We have carefully checked the formula section of the article to ensure that all parameters in the formula are stated and elaborated.

Point 4: Conclusion. The final chapter should be expanded, and practical implications of your research should be elaborated in more detail. It would be meaningful if you would discuss the results by suggesting how they are relevant to further sector development and how your research could be used in practice.

  1. This segment should be enhanced with additional outcomes and comparisons with other studies in the field. Are your results aligned with those of other studies in the field? Also, the benefits of the proposed model are not clear enough and should be described more concisely.

Response 4: Thank you. We have extended the conclusion, expounded the practical significance of the research in more detail, and put forward relevant policy suggestions. See Section 7 for details.

Reviewer 2 Report

Title: Research on Carbon Emission Prediction Model of Thermal Power Plant Based on SSA-LSTM Algorithm with Boiler Feed Water Influence Factors

Comments:

1.       The abstract is not well written. There is a need to revise with explicit contents of the abstract, i.e., the main issue, methods, results, and implication. The author(s) should provide a precise and focused abstract.

2.       Abbreviations should be avoided in keywords.

3.       As a suggestion for improvement, the author(s) should not use the exact Keywords like Paper Title. It is encouraged to use different keywords which are not in the Paper title. It will enhance paper searchability after publication.

4.       Introduction and literature review need to be divided into two separate parts.

5.       The Introduction need to be rewritten. The Introduction should highlight the relevance of the topic, the novelty of the results, the importance of policy implications, the sample's choice, the methodology's appropriateness, the data used, the contribution to the literature, and the limitations of the study.

6.       Literature review is partial and incomplete, and some recent and relevant contributions should be cited and discussed.    

7.       The next line “Where” of all formulas should be changed to “where”.

8.       (1)-(6) in “4.2.2 significance analysis of direct influencing factors of carbon emissions from coal-fired power plants” need to be changed to English format.

9.       The format of references is not uniform. It is suggested to modify the article format according to the requirements of the journal.

10.   Please add policy suggestions.

11.   The English of this manuscript should be improved furthermore. It is better to invite a native speaker to edit the whole paper.

Final views: major revision

Author Response

Response to Reviewer 2 Comments

Point 1: The abstract is not well written. There is a need to revise with explicit contents of the abstract, i.e., the main issue, methods, results, and implication. The author(s) should provide a precise and focused abstract.

Response 1: Thank you. We have revised the abstract according to your suggestion.

Point 2:   Abbreviations should be avoided in keywords.

Response 2: Thank you, we have already described the abbreviation keywords in the abstract.

Point 3:  As a suggestion for improvement, the author(s) should not use the exact Keywords like Paper Title. It is encouraged to use different keywords which are not in the Paper title. It will enhance paper searchability after publication.

Response 3: Thank you. We have modified the key words according to your suggestion.

Point 4: Introduction and literature review need to be divided into two separate parts.

Response 4: Thank you. We have divided the introduction and literature review into two separate chapters according to your suggestion.

Point 5: The Introduction need to be rewritten. The Introduction should highlight the relevance of the topic, the novelty of the results, the importance of policy implications, the sample's choice, the methodology's appropriateness, the data used, the contribution to the literature, and the limitations of the study.

Response 5: Thank you. We have rewritten the introduction as you suggested. The relevance of the article, the novelty of the results, the importance of the policy implications, the selection of the sample, the appropriateness of the methodology, the data sources, the contribution to the literature, and the limitations of the study are highlighted.

Point 6: Literature review is partial and incomplete, and some recent and relevant contributions should be cited and discussed.

Response 6: Thank you. We have supplemented the literature review according to your comments.

Point 7: The next line “Where” of all formulas should be changed to “where”.

Response 7: Thank you. We have completed the replacement as you suggested.

Point 8:  (1)-(6) in “4.2.2 significance analysis of direct influencing factors of carbon emissions from coal-fired power plants” need to be changed to English format.

Response 8: Thank you. We have adjusted the format in 4.2.2 according to your suggestion.

Point 9:  The format of references is not uniform. It is suggested to modify the article format according to the requirements of the journal.

Response 9: Thank you. We have adjusted the reference format according to your comments.

Point 10: Please add policy suggestions

Response 10: Thank you. We have added policy recommendations in Chapter 7.

Point 11: The English of this manuscript should be improved furthermore. It is better to invite a native speaker to edit the whole paper.

Response 11: Thank you, we have invited native speakers to polish the article.

Round 2

Reviewer 1 Report

The authors improved the paper, which is in much better shape than original. There are, however, a few remaining issues that need to be addressed.

1)     Literature review is an important part of any scientific paper. The authors did provide a number of references.  However, a more thorough literature review should be conducted and the specific scientific gap outlined.

2)     Methodology. Developing a suitable methodology is crucial. The authors spent too little time in describing their research principles. The importance of having a more detailed description of the method is twofold: first, it should lead to reproducibility of the results and second, it should outline the expansion of knowledge required for the method to be considered a novel contribution in the field. At present, the reproducibility of results would be very difficult due to the high-level explanations of the mathematical framework. For instance, breaking down the very first objective function seems to be an impossible task: i.e. what are “re parameters” and how do you define “leaf nodes”?

a.      The authors also utilize a software simulation tool. To what extent is the method applied a novel contribution?

3)     Conclusion. The final chapter should be expanded, and practical implications of your research should be elaborated in more detail. It would be meaningful if you would discuss the results by suggesting how they are relevant to further sector development and how your research could be used in practice.

a.      Discussion. This segment should be enhanced with additional outcomes and comparisons with other studies in the field. Are your results aligned with those of other studies in the field? Also, the benefits of the proposed model are not clear enough and should be described more concisely.

Author Response

Point 1: Literature review is an important part of any scientific paper. The authors did provide a number of references.  However, a more thorough literature review should be conducted and the specific scientific gap outlined.

Response 1: Thank you. We have thoroughly revised the literature review and compared the machine learning algorithm with the traditional gray scale prediction algorithm. It highlights the superiority of the model established in this paper.

Point 2: Methodology. Developing a suitable methodology is crucial. The authors spent too little time in describing their research principles. The importance of having a more detailed description of the method is twofold: first, it should lead to reproducibility of the results and second, it should outline the expansion of knowledge required for the method to be considered a novel contribution in the field. At present, the reproducibility of results would be very difficult due to the high-level explanations of the mathematical framework. For instance, breaking down the very first objective function seems to be an impossible task: i.e. what are “re parameters” and how do you define “leaf nodes”?

a.The authors also utilize a software simulation tool. To what extent is the method applied a novel contribution?

Response 2: Thank you. In Chapter 6, we used different algorithms to process two different data sets respectively to verify the robustness of the model and ensure that the model can be reproduced. In this paper, software simulation tools are used to analyze various factors affecting carbon emissions. This method can more clearly and intuitively reflect the influence trend of the changes of various factors on the results of carbon emissions, so as to screen out the factors with significant influence.

Point 3: Conclusion. The final chapter should be expanded, and practical implications of your research should be elaborated in more detail. It would be meaningful if you would discuss the results by suggesting how they are relevant to further sector development and how your research could be used in practice.

a.Discussion. This segment should be enhanced with additional outcomes and comparisons with other studies in the field. Are your results aligned with those of other studies in the field? Also, the benefits of the proposed model are not clear enough and should be described more concisely.

Response 3: Thank you. We have revised the conclusions of this paper. We have made a more concise description of the advantages of the model in 7.1, and the comparison of the results highlights that the performance of the model established in this paper is superior to other research results in this field. Finally, policy recommendations and application scenarios are added in 7.2.

Reviewer 2 Report

The author has carried on the detailed revision and the supplement to the manuscript, the logic is clear, the innovation point is bright, the content is substantial, the realistic significance is strong. Therefore, I agree to accept this manuscript.

Author Response

Thank you for your recognition of our research. In future studies, based on the research paradigm proposed in this paper, we will further study the carbon peaking and carbon neutralization of China's power industry through novel intelligent computing methods, explore the specific process of carbon peaking and carbon neutralization, and put forward corresponding solutions for reference.

Round 3

Reviewer 1 Report

The authors revised the paper according to the remarks made.